# Two halves are less than the whole: Evidence of a length bisection bias in fish (*Poecilia reticulata*)

**Maria Santacà****[1]\*, Christian Agrillo[1,2]**

**1** Department of General Psychology, University of Padova, Padova, Italy, **2** Padua Neuroscience Center, University of Padova, Padova, Italy

\* santacamaria@gmail.com

**Data Availability Statement:** The data are available in the paper (Table 1).

**Funding:** The present work was carried out within the scope of the research program 'Dipartimenti di Eccellenza' (art.1, commi 314-337, legge 232/

## Abstract

The horizontal-vertical (HV) illusion is characterized by a tendency to overestimate the length of vertically-arranged objects. Comparative research is primarily confined to primates, a range of species that, although arboreal, often explore their environment moving along the horizontal axis. Such behaviour may have led to the development of asymmetrical perceptual mechanisms to make relative size judgments of objects placed vertically and horizontally. We observed the susceptibility to the HV illusion in fish, whose ability to swim along the horizontal and vertical plane permits them to scan objects' size equally on both axes. Guppies (*Poecilia reticulata*) were trained to select the longer orange line to receive a food reward. In the test phase, two arrays, containing two same-sized lines were presented, one horizontally and the other vertically. Black lines were also included in each pattern to generate the perception of an inverted T-shape (where a horizontal line is bisected by a vertical one) or an L-shape (no bisection). No bias was observed in the L-shape, which supports the idea of differential perceptual mechanisms for primates and fish. In the inverted T-shape, guppies estimated the bisected line as shorter, providing the first evidence of a length bisection bias in a fish species.

## Introduction

The horizontal-vertical (HV) illusion is one of the most studied distortion illusions [e.g. 1, 2, 3]. In its classical configuration, it is represented as an inverted T: although the length of the horizontal line is identical to that of the vertical line, the latter appears longer to human observers (Fig 1). To explain this phenomenon, three main hypotheses have been advanced. The first one–hereafter called 'visual field shape' hypothesis—is based on assimilation / contrast effects associated with the oval shape of our binocular visual field [1, 4]. The binocular visual field is a horizontally-oriented ellipse. The ends of the vertical line are likely to be found near the boundary of the visual field, which allows the vertical line to assimilate the perceptual properties of the vertical visual field, appearing slightly longer. By contrast, the ends of the horizontal line are likely to be found far from the boundary of the horizontal visual field. This

2016). This work was supported by PRIN 2015 (grant number: 2015FFATB7) from MIUR and by 'STARS@unipd' (ANIM_ILLUS) grant from the University of Padova to C.A.

contrast enables us to perceive the horizontal line as smaller. Taken together, both effects have the potential to generate the HV illusion.

An alternative hypothesis is known as the 'inappropriate size-scaling hypothesis' [5, 6, 7]. This is based on the common experience of perceiving three-dimensional objects through two-dimensional stimuli and is conceived as being at the root of several illusory phenomena [reviewed in 8]. In the case of the HV pattern, the vertical line may be perceived behind the horizontal line and/or receding into the distance. We are equipped with size-constancy mechanisms that permit us to compensate for the extent to which an object's retinal size decreases as its distance increases. Therefore, if two lines occupy the same retinal space, but one is perceived as being below the other or receding into the distance, this line is typically perceived as being longer than the horizontal one, which appears to lie in front of us. The two hypotheses are not mutually exclusive and both the assimilation/contrast effects associated with the visual field shape and size-constancy mechanisms may operate in the same direction to elicit the experience of a subjective phenomenon called 'anisotropy of the perceived space', namely the asymmetrical perception of size in the vertical and horizontal axis.

The third hypothesis is associated with the 'length bisection bias' [9, 10]. When an object is divided in two parts, it appears shorter than it does when there is no division. In the case of the HV pattern, the horizontal line is bisected by the vertical one, a fact that can make the former appear to be smaller. Consistent with this hypothesis, it was found that the magnitude of the illusory perception in the 'inverted-T' version is larger than that reported in the 'L' version, where no line is bisected [10]. However, the length bisection bias alone cannot explain the HV illusion, because we perceive the vertical line as being longer than the horizontal one in the 'L' version.

There is evidence that non-human animals are also susceptible to the HV illusion. With the exception of a study of hens [11], all the studies are confined to non-human primates. Using

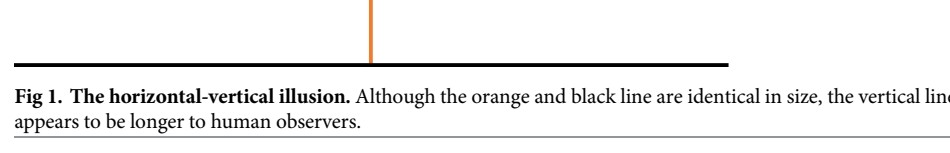

**Fig 1. The horizontal-vertical illusion.** Although the orange and black line are identical in size, the vertical line appears to be longer to human observers.

an operant conditioning procedure, Domingues trained three rhesus monkeys, one mangabey and two capuchin monkeys to select the smaller line between two alternative lines [12]. In the presence of the HV pattern, most of the monkeys proved to be susceptible to this distortion illusion. Stumptail monkeys were also found to perceive the HV illusion [13]. Although we acknowledge the importance of studying the HV illusion in non-human primates, the ecology of the species is not drastically different among primates, a fact that limits our comprehension of the universality of the anisotropy of the perceived space among vertebrates. As far as we are aware, this phenomenon may be confined to primates only, or more generally, may be limited to species that are differentially exposed to the possibility of exploring objects in the vertical and horizontal space. Indeed, even though non-human primates are also arboreal, they usually spend a considerable amount of time on the ground of the forest. Comparing objects' size, when the objects are arranged on two different axes may be difficult for species navigating in the space through the horizontal axis, and the possibility exists that neurocognitive systems of such species have been selected to adopt indirect perceptual strategies (e.g., based on assimilation/contrast effect, or size-constancy) to interpret the information coming to the retina about objects found vertically and horizontally.

Fish may play an important role in shedding light on this issue. Fish represent the largest vertebrate group, because half of vertebrate species are teleost fishes [14]. As is well-known, fish are not merely capable of swimming forward and backward in the horizontal axis, but are also capable of swimming upwards and downwards, a fact that may have led to a more balanced capacity to compare the relative size of objects presented in the vertical and horizontal axes. Recent studies showed that fish are susceptible to different visual illusions, including the Müller Lyer illusion, Ebbinghaus illusion, Delboeuf illusion, brightness illusion, rotating snake illusion and the Solitaire illusion [reviewed in 15]. However, no study has investigated the HV in aquatic species. In the present study, we investigated the susceptibility to the HV illusion in the guppy (*Poecilia reticulata*), a freshwater teleost fish commonly used in cognitive/perceptual studies [16, 17, 18]. We adopted an operant conditioning procedure that consisted of training subjects to select the longer orange line between two alternative orange lines. Two arrays were presented in the training phase, in which the orange line was presented either vertically or horizontally. In each array, a black line was also presented, to generate an L shape or an inverted-T shape. Guppies had to reach the longer orange line to receive a food reward. As soon as guppies achieved the learning criterion, they were presented with the illusory patterns consisting in two same-sized orange lines presented in two different arrays: in one array, the orange line was presented vertically; in the other, it was presented horizontally. Again, black lines, perpendicular to the orange lines, were also included in each array to generate a L shape or an inverted-T shape. If guppies were susceptible to the illusion, they were expected to select the array presenting the orange line vertically. This may imply that the anisotropy of the perceived space also exists in species living under the surface of the water. We predict that, as fish navigate equally well in the horizontal and vertical space, they may be less susceptible (if they are susceptible at all) to the HV illusion than primates. In addition, the comparison of the guppies' performance, in the presence of the L-shape or inverted T-shape, reveals to us whether guppies are susceptible to the length bisection bias, a bias never reported in fish species. In the absence of specific literature on the issue, we were able to make no prediction with respect to this perceptual bias.

## Materials and methods

### Ethics statements

We followed all applicable international, national, and/or institutional guidelines for the care and use of animals (Italy, D.L. 4 Marzo 2014, n. 26). The study was performed in

accordance with the ethical standards of the institution at which the study was conducted and was approved by the Ethical Committee of the Università di Padova (Protocol n. 32/2019).

## Subjects

The experimental design included a sample size of 12 tested subjects, a sample size previously adopted in similar studies in the same species [e.g. 19, 20]. However, as is true of any training procedure, some animals had to be discharged, because they stopped participating in the experiment (4 subjects) or failed to achieve any of the learning criteria necessary to pass onto the test phase (6 subjects). Therefore, the total sample size included 22 adult and naïve female guppies. The guppies belonged to an ornamental strain known as the 'snakeskin cobra green' strain, which is regularly bred in our laboratory in the Department of General Psychology of the University of Padova (Italy). The guppies were housed in several 80-L tanks enriched with a gravel bottom and plants (*Hygrophila corymbosa* and *Taxiphyllum barbieri*). The water was maintained at a temperature of 26 ± 1˚C and was aerated by means of a biomechanical filter. A 12:12 h light:dark photoperiod was ensured by a 30-W phytostimulant lamp placed above each tank. Guppies were fed with commercial food flakes (Aquatropical, Padovan©) in the morning and with live brine shrimp nauplii (*Artemia salina*) in the afternoon. The experimental subjects spontaneously participated and none appeared to be experiencing stress during the experiment. Once the subjects finished the experiment, they were moved to apposite maintaining tanks and kept for breeding.

## Apparatus

The experiment was performed in 12 identical glass tanks (20 × 50 × 32 cm), each filled with 28 L of water (Fig 2), and previously used in similar experiments in the same species [e.g. 19, 20]. Two transparent hourglass-shaped acetate sheets (10 × 6 × 32 cm) divided each tank into four compartments: two lateral compartments, a frontal compartment, and a posterior compartment. In each lateral compartment, a mirror (28 × 5 cm), a plant and two immature conspecifics were inserted. The plants were necessary to replicate a natural environment, whereas the mirrors and the conspecifics prevented any stress due to the social isolation [21]. Each tank was further enriched with loose gravel and each side was covered externally with a green plastic sheet to avoid any external disruptive influence. Identically to the maintaining tanks, the water temperature was maintained at 26 ± 1˚C and a 30 W phytostimulant lamp illuminated the experimental tanks, according to a 12:12 h light:dark photoperiod.

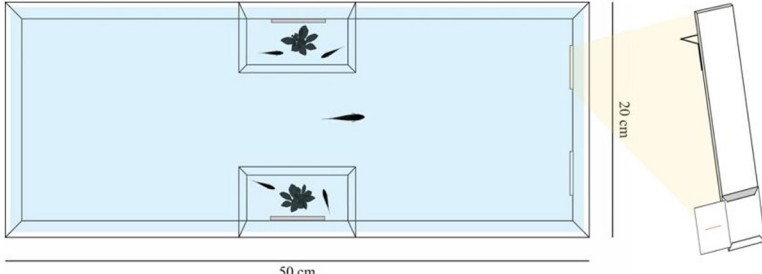

**Fig 2. Experimental setup.** An aerial representation of the experimental tank and a three-dimensional representation of the panel used to present the stimuli to the subjects.

## Procedure and stimuli

The experimental procedure consisted of three phases: a pre-training phase, a training phase and a test phase.

**Pre-training phase.** The pre-training phase lasted for two consecutive days, during which the subjects were able to familiarize themselves with the experimental tanks. During this phase, each subject was presented with a target orange line, which was printed on a white laminated card (3 × 3 cm). The colour orange was chosen because guppies are spontaneously attracted to orange stimuli [22]. To present the stimuli to the subjects, each card was affixed to the end of a transparent panel (3.5 × 15 cm) that was able to be fixed on the wall of the tank by means of an L-shaped blocker. On the first day, each subject was able to see the orange target line (presented horizontally or vertically) eight times, divided into four trials in the morning session and four trials in the afternoon session; a 90-minute interval separated the two sessions. Only if the subject approached the stimulus, a food reward, which consisted in a drop of live brine shrimps delivered by a Pasteur pipette, was provided. In each session, it was possible to perform a new trial after an interval of 15 minutes following the previous trial. On the second day, each subject underwent 12 trials, six in the morning session and six in the afternoon session with a 90-minute interval between the two sessions. Each trial consisted of the presentation of two lines that differed in length by a ratio of 0.67. In fact, the longer line was 1.75 cm in length whereas the short line was 1.17 cm in length. This ratio was previously adopted in similar investigations in the same species that had already demonstrated to discriminate it [e.g. 19, 20]. Due to the guppies' demonstrably high attraction to the orange colour, we expected the subjects to approach the longer line. In the case in which the subjects approached the shorter line, the trial continued until they approached the longer one, and only in that case was the food reward provided, thereby adopting a correction procedure. The horizontal/vertical and left/right position of the longer line was counterbalanced across the trials such as the short side of the tank in which the stimuli were presented.

**Training phase.** The training phase lasted a maximum of 10 consecutive days, during which the subjects performed 12 trials with the same procedure employed on the second day of the pre-training phase. Four different types of trials were arranged: 'Horizontal Inverted-T Control', 'Vertical Inverted-T Control', 'Horizontal L Control' and 'Vertical L Control' (Fig 3). The horizontal controls (Horizontal Inverted-T Control and Horizontal L Control) featured two different-length orange lines which were arranged horizontally; both stimuli also featured the inducers, namely two same-length black lines which were placed vertically (1.75 cm in length; Fig 3A and 3D). The vertical controls (Vertical Inverted-T Control and Vertical L Control) instead featured two different-length target lines, which were arranged vertically, and the inducers, which were arranged horizontally (Fig 3B and 3E). The difference between the T-shaped controls (Horizontal Inverted-T Control and Vertical Inverted-T Control) and the L-shaped controls (Horizontal L Control and Vertical L Control) consisted of the position of the inducers, relative to the target lines. In fact, in the T-shaped controls, the horizontal line was bisected by the vertical one, whereas in the L-shaped controls, no line was bisected. In all four trial types, the ratio between the target lines was 0.67 (the longer line was 1.75 cm in length, whereas the shorter line was 1.17 cm in length). In all control trials of the training phase, if the subjects approached the longer target line first, they were given the food reward. On the contrary, if they approached the shorter target line first, no correction was allowed and no food reward was provided. Each day, it was possible for each subject to face each type of trials three times. The left/right position of the longer target line and the short side of the tank, in which the stimuli were presented, were counterbalanced over the trials. Only the subjects that met one of two learning criteria were able to pass on to the subsequent test phase. One learning

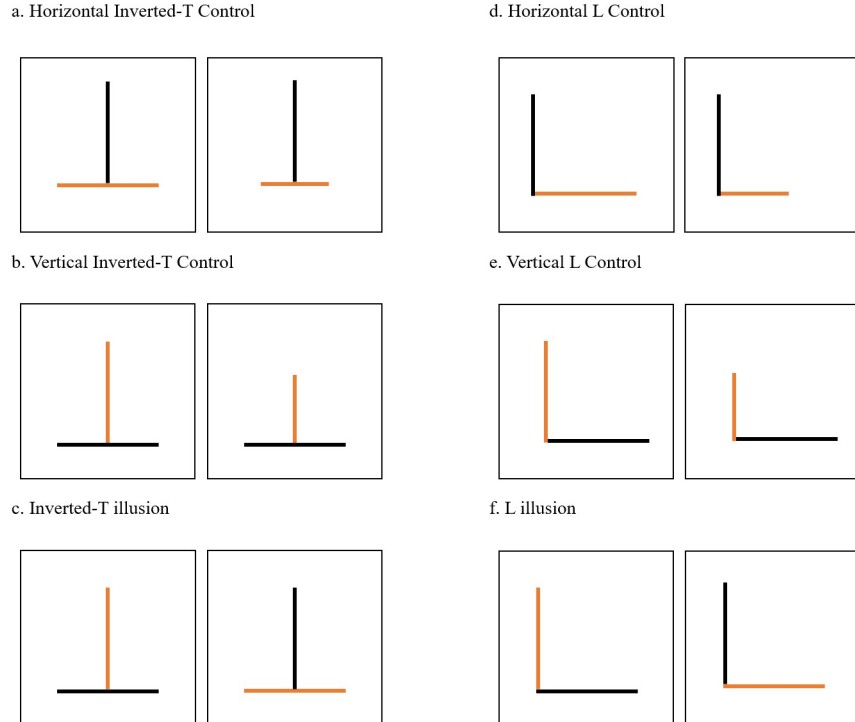

a. Horizontal Inverted-T Control

d. Horizontal L Control

b. Vertical Inverted-T Control

e. Vertical L Control

c. Inverted-T illusion

f. L illusion

**Fig 3. Experimental stimuli.** Two white cards containing two different- or same-length target lines were presented: (a) Horizontal Inverted-T Control, (b) Vertical Inverted-T Control, (c) Inverted-T illusion, (d) Horizontal L Control, (e) Vertical L Control and (f) L illusion.

criterion consisted of a statistically significant performance of at least 75% correct choices in the trials across two consecutive days (binomial test: 18/24, $p < 0.05$). The other learning criterion was defined as a statistically significant performance of 72 correct choices across the 120 trials (binomial test: $p < 0.05$). A similar set of dual criteria was previously adopted in a similar investigation in the same species [20].

**Test phase.** The test phase lasted 10 consecutive days during which the subjects participated in 12 trials, using the same procedure as on the second day of the pre-training phase and of the training phase. Each day, 8 out of 12 trials consisted of the same control trials as the training phase; in fact, subjects faced each type of control trial twice every day, which came to a total of 20 trials for each type. In a manner identical to the training-phase procedure, we administered the food reward in response to correct choices in these types of trials. Alternative to these control trials, it was possible for the subjects to face two different types of HV illusory trials: the inverted-T illusion and the L illusion (Fig 3C and 3F). In both illusory trials, two same-length target lines were presented in two different arrays: in one array, the orange line was presented vertically, whereas, in the other, it was presented horizontally. Black inducers, perpendicular to the target lines, were also included in each array to generate an L shape or an inverted-T shape. In both types of illusory trials, guppies were expected to choose the target line that they perceived as being longer. To avoid any learning bias, we did not provide any food reward irrespective of the chosen stimulus, in both types of illusory trials. The six different types of trials were presented, according to a predetermined pseudorandom sequence that was different for each subject. The left/right position of the longer target line (or of the vertical line, which was the one perceived as being longer by human observers, in both illusory trials) and the short side of the tank, in which the stimuli were presented, were counterbalanced over the trials.

## Statistical analyses

Data were analysed in R version 3.5.2 (the R Foundation for Statistical Computing, Vienna, Austria, http://www.rproject.org). Two-proportions z-tests ('prop.test' function) were performed to compare the choices for the longer line in the control trials or for the vertical line (the one perceived as being longer by humans) in both types of illusory trials (chance level = 0.50). A generalised mixed-effects model for binomial distributions (GLMM, 'glmer' function of the 'lme4' R package) was conducted to compare the performance across all six types of trials using the choices of the longer line in the four types of control trials and of the vertical line in both types of illusory trials. In the same model, we also checked for any eventual effect of the day to ensure that the performances of the 12 subjects remained stable across all 10 days. In case of a significant effect of the type of trial or of the day, all pairwise comparisons were conducted with the Tukey post hoc test [23]. To assess eventual inter-individual differences, we conducted another generalised mixed-effects model. Cohen's $d$ ('cohensD' function of the 'lsr' package) was conducted as an effect size statistic: a $d$ of 0.2 represents a small effect size, a $d$ of 0.5 represents a medium effect size, and a $d$ of 0.8 represents a large effect size [24].

# Results

## Training phase

Eleven out of 12 guppies passed on to the test phase after meeting the learning criterion of 70% correct choices in 24 trials over two consecutive days. On average, the 11 guppies performed 53.45 trials (SD = 14.56) before passing on to the test phase. The remaining subject achieved a significant frequency of correct choices over the total of 120 trials (binomial test: 74/120, $p < 0.05$).

## Test phase

The guppies significantly selected more than chance the longer line in both T-shaped controls: the Horizontal Inverted-T Control (mean: 0.692, 95% CI [0.629, 0.749], $p < 0.001$, $d = 1.593$; Fig 4) and the Vertical Inverted-T Control (mean: 0.658, 95% CI [0.595, 0.718], $p < 0.001$, $d = 1.192$; Fig 4). Also in both L-shaped controls, the guppies significantly selected the longer line more than chance: the Horizontal L Control (mean: 0.621, 95% CI [0.556, 0.682], $p < 0.001$, $d = 1.397$; Fig 4) and the Vertical L Control (mean: 0.733, 95% CI [0.673, 0.788], $p < 0.001$, $d = 3.251$; Fig 4). In the illusory trials, the guppies demonstrated a significant preference for the vertical line in the Inverted-T Illusion (mean: 0.608, 95% CI [0.543, 0.670], $p < 0.001$, $d = 1.366$; Fig 4) whereas no preference bias was exhibited in the L Illusion (mean: 0.517, 95% CI [0.451, 0.581], $p = 0.652$, $d = 0.162$; Fig 4). This suggests that the guppies were susceptible to the HV illusion but only in the classical T-inverted configuration.

The GLMM showed that the performance of the subjects was stable across the total of 10 test days ($\chi^2_9 = 8.590$, $p = 0.476$) and that it varied as a function of the trial type ($\chi^2_5 = 21.144$, $p < 0.001$). The day × type of trial interaction was not significant ($\chi^2_{45} = 34.210$, $p = 0.879$). The Tukey post hoc test revealed no significant difference among the four types of control trials: Horizontal Inverted-T Control, Vertical Inverted-T Control, Horizontal L Control and Vertical L Control (all $p$-values > 0.381). A consideration of the Inverted-T Illusion revealed no significant difference considering the T-shaped controls (both $p$-values > 0.737); rather, a consideration of the L Illusion revealed a significant difference only considering the Vertical L Control ($p < 0.001$; L illusion–Horizontal L Control: $p = 0.117$). A comparison of the performances in the two types of illusory trials yielded results that barely attained the significance threshold ($p = 0.056$). The other GLMM revealed neither a significant difference between the

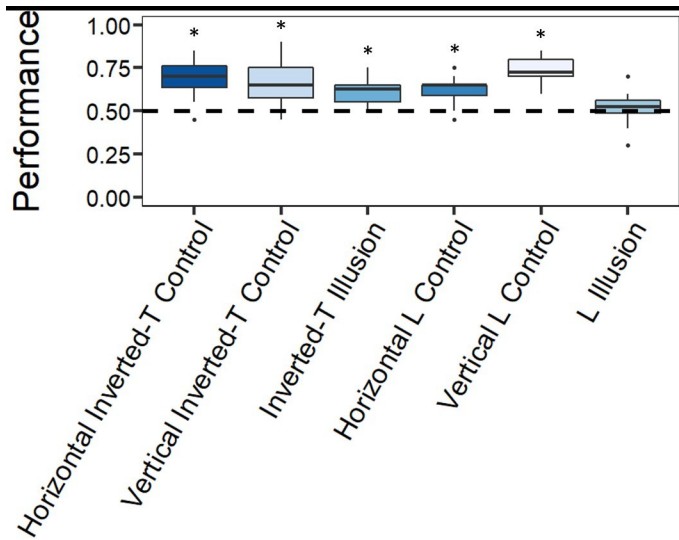

Type of trials

**Fig 4. Results.** The Y-axis refers to the proportion of choices for the longer target line in the four types of control trials, and the proportion of choices for vertical line in both illusory trials. The asterisk (*) denotes a significant departure from chance level ($p < 0.05$).

subjects ($\chi^2_{11} = 9.767$, $p = 0.552$) nor a significant interaction between the subject × type of trial ($\chi^2_{33} = 49.079$, $p = 0.699$), which indicates that no inter-individual variation was detectable. The individual data of the test phase are shown in Table 1.

## Discussion

We aimed to test whether a species distantly related to humans, such as guppies, is susceptible to the HV illusion to shed light on the evolution of perceptual mechanisms underlying size estimation of vertebrates. In particular, we predicted that, because fish equally explore the environment horizontally and vertically, they may have developed perceptual systems capable

**Table 1. Individual data.** Individual data of the test phase (Horizontal Inverted-T Control, Vertical Inverted-T Control, Horizontal L Control, Vertical L Control, Summed Controls: frequency of choices for the longer line; Inverted-T Illusion, L Illusion: frequency of choices for the vertical line).

| Subject | Vertical Inverted-T Control | Horizontal Inverted-T Control | Horizontal L Control | Vertical L Control | Summed Controls | Inverted-T Illusion | L Illusion |
|---|---|---|---|---|---|---|---|
| 1 | 13/20 | 17/20 | 12/20 | 12/20 | 54/80 | 11/20 | 11/20 |
| 2 | 16/20 | 15/20 | 13/20 | 16/20 | 60/80 | 14/20 | 9/20 |
| 3 | 15/20 | 17/20 | 13/20 | 14/20 | 59/80 | 13/20 | 14/20 |
| 4 | 13/20 | 16/20 | 14/20 | 16/20 | 59/80 | 13/20 | 8/20 |
| 5 | 10/20 | 11/20 | 13/20 | 15/20 | 49/80 | 13/20 | 11/20 |
| 6 | 12/20 | 14/20 | 15/20 | 15/20 | 56/80 | 15/20 | 10/20 |
| 7 | 15/20 | 12/20 | 13/20 | 14/20 | 54/80 | 10/20 | 12/20 |
| 8 | 18/20 | 13/20 | 10/20 | 14/20 | 55/80 | 10/20 | 10/20 |
| 9 | 14/20 | 13/20 | 12/20 | 14/20 | 53/80 | 12/20 | 10/20 |
| 10 | 10/20 | 14/20 | 13/20 | 13/20 | 50/80 | 11/20 | 12/20 |
| 11 | 13/20 | 9/20 | 9/20 | 17/20 | 48/80 | 11/20 | 11/20 |
| 12 | 9/20 | 15/20 | 11/20 | 16/20 | 51/80 | 13/20 | 6/20 |

finely matching objects' size in the two axes, leading to a smaller (if not absent) susceptibility to the HV illusion.

Our findings support this hypothesis. In the presence of two same-sized orange lines, arranged in two L-shapes (one in the horizontal space, and one in the vertical space) guppies did not select either of them more than chance. The lack of bias in the L-shaped pattern fits with neither the predictions of the 'visual field shape' hypothesis [1] nor the 'inappropriate size-scaling hypothesis' [5]. In both cases, a significant bias for selecting the longer line was expected. Therefore, there is no statistical argumentation to support the claim that guppies experience the anisotropy of the vertical space reported in humans and non-human primates [3, 12]. On the contrary, fish exhibited a perceptual bias in the presence of the inverted-T pattern, as they selected the vertical (bisecting) line more than chance. This aligns with the length bisection bias hypothesis [9, 10]. To our knowledge, this represents the first evidence of this type of perceptual bias in a fish species. The performance in the presence of the L illusion and inverted-T illusion was stable across the days, which demonstrated that no learning in these non-reinforced test trials unduly obscured a fish's genuine response to the illusory patterns.

Previous studies have reported interesting similarities between humans and guppies, with regard to susceptibility to visual illusions. Guppies perceive the rotating snake illusion [25], the brightness illusion [26] and the Solitaire illusion [27], which suggests that humans and guppies have similar perceptual mechanisms underlying motion, brightness and numerosity. The picture related to size estimation is more complex: recently guppies were found to perceive the Müller-Lyer illusion in the same way as humans [20] but another study suggested a reverse perception of the Delboeuf illusion [19]. Both illusions are two well-known distortion illusions. The Müller-Lyer illusion consists in the overestimation of the perceived relative length a line when it has inwards-pointing arrowheads. The Delboeuf illusion consists instead in the overestimation of the perceived relative size of a circle when it is presented on a small context (i.e. a circumference). In the latter case, guppies showed to overestimate the size of a circle but when it is presented in a large circumference suggesting a reverse perception of the Delboeuf illusion compared to humans [19]. The authors advanced the hypothesis that guppies may have been only minimally susceptible to contrast effects [19]. In fact, in humans, the Delboeuf illusion is known to be due to a combination of assimilation and contrast effects [19]. When a circumference is close to circle, then the circumference is meant to assimilate, leading to a perception of circle as larger; whereas when the circumference is larger and, as a consequence, far from the circle, then the circumference is meant to contrast, leading to an underestimation of the circle in size. Assimilation and contrast effects seem to operate also in the perception of the HV illusion: the ends of the vertical line are supposed to be closer to the boundary of the visual field, leading to an assimilation of its length to the boundary edges of the visual field. Instead, the ends of the horizontal line are supposed to be far from the boundary of the visual field, eliciting a contrast effect [28, 29]. A reduced susceptibility to contrast effects, found in the Delboeuf illusion study [19], may have impacted the behaviour exhibited with the L-shape, thereby reducing underestimation of the horizontal line and nulling the illusory effect. These effects, instead, are not supposed to play a role in the length bisection bias.

In the control (reinforced) trials that transpired in the presence of lines that physically differ from each other, guppies showed an overall ability to identify the longer one in both visual arrays (L-shape or inverted T-shape) with no difference across the conditions. This was a necessary condition to establish whether subjects were actually trying to select the subjectively longer line in the context of illusory trials. That guppies proved capable of a 0.67 size ratio is consistent with previous literature, which investigated relative size judgments of food items [30] and two-dimensional figures of the same species [19].

In conclusion, we found that guppies overestimated the length of a bisecting line, relative to the length of the bisected one, which is consistent with the results found in literature on humans. No perceptual biases were reported as being a function of the position of the line in the vertical or horizontal space, thereby excluding the possibility that the shape of the visual field and/or size constancy mechanisms may have played an important role in guppies' length estimation. Future studies are necessary before it is possible to draw firm conclusions concerning the lack of the anisotropy of the perceived space in fish. In the absence of further investigation, we must be open to the possibility that primates and fish have distinctive perceptual mechanisms underlying size estimation in the horizontal and vertical space.

## Acknowledgments

We thank Mara Zebele and Giada Alessi for their help in testing the subjects.

## Author Contributions

**Conceptualization:** Maria Santacà, Christian Agrillo.

**Data curation:** Maria Santacà.

**Formal analysis:** Maria Santacà.

**Funding acquisition:** Christian Agrillo.

**Investigation:** Maria Santacà.

**Methodology:** Maria Santacà, Christian Agrillo.

**Project administration:** Christian Agrillo.

**Resources:** Maria Santacà, Christian Agrillo.

**Supervision:** Christian Agrillo.

**Validation:** Maria Santacà, Christian Agrillo.

**Visualization:** Maria Santacà, Christian Agrillo.

**Writing – original draft:** Maria Santacà, Christian Agrillo.

**Writing – review & editing:** Maria Santacà, Christian Agrillo.

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
