## [Decision Letter · Decision Letter 0]

15 Apr 2020

PONE-D-20-07425

Two halves are less that the whole: evidence of a length bisection bias in fish (Poecilia reticulata)

PLOS ONE

Dear Mrs. Santacà,

Thank you for submitting your manuscript to PLOS ONE. After careful consideration, we feel that it has merit but does not fully meet PLOS ONE’s publication criteria as it currently stands. Therefore, we invite you to submit a revised version of the manuscript that addresses the points raised during the review process.

So far I have had the manuscript reviewed by one expert on animal illusion perception. This reviewer supports the publication, but also raises a number of important revisions. I have read and agree with R1. In particular, I think you must more fully review animal literature on illusions, and hence reduce claims of this study being first evidence of a length bisection bias in an animal species. I also think there is a broader literature on spatial illusions like the  Delboeuf illusion in animals that could be discussed. Please carefully address each point. I would also like that you make raw data available (see PLOS policies). If you can make all revisions in a careful way, I may proceed on the basis of the advice of the one expert reviewer.

We would appreciate receiving your revised manuscript by May 30 2020 11:59PM. To enhance the reproducibility of your results, we recommend that if applicable you deposit your laboratory protocols in protocols.io, where a protocol can be assigned its own identifier (DOI) such that it can be cited independently in the future. For instructions see: http://journals.plos.org/plosone/s/submission-guidelines#loc-laboratory-protocols

We look forward to receiving your revised manuscript.

Kind regards,

Adrian G Dyer, Ph.D.

Academic Editor

PLOS ONE

Journal Requirements:

2. Please update the word "that" in the title to "than" for grammatical correctness.

4. Your ethics statement must appear in the Methods section of your manuscript. If your ethics statement is written in any section besides the Methods, please move it to the Methods section and delete it from any other section. Please also ensure that your ethics statement is included in your manuscript, as the ethics section of your online submission will not be published alongside your manuscript.

Reviewers' comments:

Reviewer's Responses to Questions

**Comments to the Author**

1. Is the manuscript technically sound, and do the data support the conclusions?

Reviewer #1: Yes

2. Has the statistical analysis been performed appropriately and rigorously? 

Reviewer #1: Yes

3. Have the authors made all data underlying the findings in their manuscript fully available?

Reviewer #1: No

4. Is the manuscript presented in an intelligible fashion and written in standard English?

Reviewer #1: Yes

5. Review Comments to the Author

Reviewer #1: The authors studied the HV visual illusion in guppies. They found that guppies were susceptible to the illusion when a line bisected the horizontal line - but this was not the case with 'L-shaped' stimuli. I would recommend this manuscript for publication with some minor revisions:

1. The authors state that there are some restrictions on data availability. Is this necessary? Why can the data not be publicly accessible?

2. The title has an grammatical error. It currently reads: "Two halves are less THAT the whole: evidence of a length bisection bias in fish (Poecilia reticulata)". I think the authors meant to write: "Two halves are less THAN the whole: evidence of a length bisection bias in fish (Poecilia reticulata)"

3. Line 15 - "explores" should be "explore".

4. I am confused by the claim the authors make multiple times (e.g. line 25-26 "... providing the first evidence of a length bisection bias in an animal species." I thought that this has been shown in non-human primates and hens? If this is the case, the lines and the other claims like it throughout the manuscript should be removed.

5. Line 144 - "breed" should be "bred".

6. Line 227 - "subjects" should be changed to "subject" as the authors state that only one guppy took the 120 trials instead of the 24 trials of the other 11 guppies.

7. Lines 258-260 - The first sentence of the discussion is messy and should be revised. The second part of the sentence does not make sense: "... that the results of which experiment have the capacity to shed light on the evolution of perceptual mechanisms underlying size estimation of vertebrates."

8. Line 258 - "is susceptible" should be changed to "are susceptible" for grammar purposes.

9. Line 270 - Change "human" to "humans".

10. Line 270 - Change "primate" to "primates".

11. 270 - delete the word "so".

12. Lines 280-284 - This is a very long sentence and should be broken up and made clearer. Currently it is confusing. The Delboeuf illusion should also be explained better for readers to understand how/why humans perceive it in a certain way and how/why guppies perceive it differently.

13.

6. PLOS authors have the option to publish the peer review history of their article (what does this mean?). If published, this will include your full peer review and any attached files.

Reviewer #1: No

---

## [Author Response · Author response to Decision Letter 0]

23 Apr 2020

A detailed response letter is attached to the submission as a word file.

---

## [Editor Report · Decision Letter 1]

28 Apr 2020

PONE-D-20-07425R1

Two halves are less than the whole: evidence of a length bisection bias in fish (Poecilia reticulata)

PLOS ONE

Dear Mrs. Santacà,

Thank you for submitting your manuscript to PLOS ONE. After careful consideration, we feel that it has merit but does not fully meet PLOS ONE’s publication criteria as it currently stands. Therefore, we invite you to submit a revised version of the manuscript that addresses the points raised during the review process.

I have carefully read the revised manuscript which I find very interesting. I have however two suggestions that I think would improve the manuscript for readers.

AGD1 Given the size of the Sup Table S1 containing the data requested by reviewer 1 (and myself to be compliant with PLOS1 policies) is not very large, I would like to suggest you just add this table to the main mainscript so that all information is at hand for readers of the manuscript. Else if you want to retain this information only as sup file I think it would be good to label that file so that when it is downloaded it is clear what the information relates to. But as stated above, if the information is placed in the main manuscript for readers and clearly labled it might be best? I will leave this decision to you.

AGD2 Regarding your revised text ["In the latter case, guppies showed not to overestimate the size of a circle presented in a small circumference. On the contrary, they demonstrated to underestimate the size of such circle and to overestimate the size of the same circle but presented in a large circumference suggesting a reverse perception of the Delboeuf illusion compared to humans [19]."]. I found this section of text not very clear. Do you mean [guppies showed evidence of not overestimating size.....]? I found the logic of these 2 sentences difficult to understand as written; can you please re write and cross check with lab members so we are confident that the general readers of PLOS will easily understand you meaning?

If you can attend to these two points I am confident I will be able to accept the manuscript.

We would appreciate receiving your revised manuscript by Jun 12 2020 11:59PM. To enhance the reproducibility of your results, we recommend that if applicable you deposit your laboratory protocols in protocols.io, where a protocol can be assigned its own identifier (DOI) such that it can be cited independently in the future. For instructions see: http://journals.plos.org/plosone/s/submission-guidelines#loc-laboratory-protocols

We look forward to receiving your revised manuscript.

Kind regards,

Adrian G Dyer, Ph.D.

Academic Editor

PLOS ONE

---

## [Author Response · Author response to Decision Letter 1]

29 Apr 2020

A detailed response letter to the comments is attached as a word file.

---

## [Editor Report · Decision Letter 2]

30 Apr 2020

Two halves are less than the whole: evidence of a length bisection bias in fish (Poecilia reticulata)

PONE-D-20-07425R2

Dear Dr. Santacà,

We are pleased to inform you that your manuscript has been judged scientifically suitable for publication and will be formally accepted for publication once it complies with all outstanding technical requirements.

With kind regards,

Adrian G Dyer, Ph.D.

Academic Editor

PLOS ONE
---

## [Editor Report · Acceptance letter]

5 May 2020

PONE-D-20-07425R2 

Two halves are less than the whole: evidence of a length bisection bias in fish (*Poecilia reticulata*) 

Dear Dr. Santacà:

I am pleased to inform you that your manuscript has been deemed suitable for publication in PLOS ONE. Congratulations! Your manuscript is now with our production department. 

With kind regards,

on behalf of

Dr. Adrian G Dyer 

Academic Editor

PLOS ONE